# Evaluation of Concomitant Orbital Floor Fractures in Patients with Head Trauma Using Conventional Head CT Scan: A Retrospective Study at a Level II Trauma Center

**DOI:** 10.3390/jcm8111852

**Published:** 2019-11-02

**Authors:** Li-Kuo Huang, Hsi-Feng Tu, Liang-De Jiang, Ying-Yuan Chen, Chih-Yuan Fu

**Affiliations:** 1Department of Radiology, National Yang-Ming University Hospital, Yi-Lan 260, Taiwan; huangcm69@yahoo.com.tw (L.-K.H.);; 2Department of Radiology, School of Medicine, National Yang-Ming University, Taipei 112, Taiwan; 3Department of Dentistry, National Yang-Ming University Hospital, Yi-Lan 260, Taiwan; 4Department of Dentistry, Dental School, National Yang-Ming University, Taipei 112, Taiwan; 5Department of Trauma and Emergency Surgery, Chang Gung Memorial Hospital, Taoyuan 333, Taiwan; 6School of Medicine, Chang Gung University, Taoyuan 333, Taiwan

**Keywords:** clear sinus sign, emergency departments, head CT, head trauma, maxillary hemosinus, orbital floor fracture, radiation exposure

## Abstract

Background: Patients with head trauma may have concomitant orbital floor fractures (OFFs). The objective of our study was to determine the specific CT findings and investigate the diagnostic performance of head CT in detecting OFFs. Methods: We analyzed 3534 head trauma patients undergoing simultaneous head and facial CT over a 3-year period. The clinical data and specific head CT findings between patients with and without OFFs were compared. Results: In our cohort, 198 patients (5.6%) had OFFs visible on CT. On head CT, orbital floor discontinuity, gas bubbles entrapped between floor fragments, inferior extraconal emphysema, and maxillary hemosinus (MHS) were more commonly observed among patients with OFFs (*p* < 0.001). The absence of MHS had a high negative predictive value (99.7%) for excluding OFFs. Among the different types of MHS, the pattern showing high-attenuation opacity mixed with mottled gas had the highest positive predictive value (69.5%) for OFFs and was the only independent predictor of OFFs after adjusting for the other CT variables in all patients with MHS. Conclusion: Head CT may serve as a first-line screening tool to detect OFFs in head trauma patients. Hence, unnecessary facial CT and additional radiation exposure may be reduced.

## 1. Introduction

As maxillofacial fractures are frequently concomitant with head trauma because of the close anatomical proximity of the facial skeleton to the cranium [1,2], orbital fractures account for a significant portion of facial trauma [3,4,5]. Furthermore, among orbital fractures, the orbital floor is the most commonly involved wall due to its thin bony structure [3,4,6]. Due to the high risk of ocular complications such as extraocular movement limitation and diplopia, in the presence of orbital floor fractures (OFFs) [7,8], early diagnosis of OFFs is necessary during the evaluation of head trauma at emergency departments.

Dedicated facial CT images provide a rapid and detailed evaluation of the bony structures and soft tissues of the orbit, and usually guide clinical decisions regarding the surgical management of orbital trauma at emergency departments [9]. For the incidence rate of 19.7% regarding concomitant orbital fractures associated with head trauma reported in one previous study [10], if there are no or only subtle fracture-related symptoms, and if clinically patients are unable to cooperate during the ophthalmologic examination, the orbital fractures may be left undetected without ordering additional facial CT scans. However, the liberal use of additional facial CT surveillance in head trauma patients without clinical signs of orbital injury will lead to another issue of unnecessary radiation exposure. 

Because head CT is the imaging modality of choice during the initial assessment of head trauma [11,12], and the orbits are usually included in its scanning range, identifying radiological features associated with OFFs on head CT may facilitate early diagnosis and minimize the number of missed fractures while avoiding unnecessary additional CT scans and associated radiation exposure. Therefore, in the current study, we aimed to evaluate the diagnostic performance of head CT in detecting OFFs and determine the radiological indicators of OFFs in head trauma patients.

## 2. Materials and Methods

### 2.1. Study Setting and Selection of Participants

This retrospective study was conducted at a secondary care hospital with a level II trauma center in Eastern Taiwan. Research ethics board approval was obtained from National Yang-Ming University Hospital (NYMUH IRB No. 2017A029).

The study cohort consisted of consecutive patients aged 20 years or older who were identified retrospectively using our picture archiving and communication system (PACS) from 1 January 2015 to 31 December 2017. Patients with head trauma who received CT scans for an evaluation for intracranial injuries were included. The clinical criteria for CT scans was a Glasgow Coma Scale (GCS) score of <13 or a minor head injury with a GCS score of 13–15, with one or more of the following findings: Headache, vomiting, older than 60 years of age, drug or alcohol intoxication, short-term memory deficit, physical evidence of supraclavicular trauma, or seizures [13,14].

### 2.2. Imaging Protocol of CT Scans

At our institution, the protocol for CT scans for head trauma patients includes both head and facial images. The head CT examinations were obtained helically without IV contrast material from the foramen magnum to the skull vertex in an axial plane paralleling the orbitomeatal line (OML). They were then reconstructed with a 5 mm slice thickness using a soft tissue kernel and a 3 mm slice thickness using a sharp kernel to evaluate the cranium. The orbit and part of the maxillary sinus were included in the scanning range of head CT. The facial CT examinations were obtained helically from the frontal sinuses through the mental symphysis and scanned parallel to the hard palate. The images were reconstructed in the axial plane at a thickness of 3 mm using a soft tissue kernel. In order to evaluate the facial bones, multiplanar CT reconstruction was routinely performed with 3 mm axial, coronal, and sagittal images using a sharp kernel. All CT studies were performed using two 64-section multiple detector CT (MDCT) scanners (Brilliance 64, Philips Medical Systems, Best, Eindhoven, The Netherlands; Somatom Definition 64 AS, Siemens Healthcare, Erlangen, Germany).

### 2.3. Image Analysis

Two board-licensed radiologists (L.D.J. and Y.Y.C.) who were blinded to the patients’ clinical information reviewed only the head CT images by means of consensus. The radiologists were permitted to manipulate the window and level of the images. The CT variables related to the cranium included intracranial hemorrhage (ICH; epidural hemorrhage, subdural hemorrhage, subarachnoid hemorrhage, and intracerebral hemorrhage), and skull fractures. The CT variables related to OFFs included orbital floor discontinuity (Figure 1), gas bubbles entrapped between the floor fragments (Figure 1), inferior extraconal emphysema (Figure 2), orbital fat herniation into the maxillary sinus (Figure 2), and ipsilateral maxillary hemosinus (MHS, Figure 3). MHS was defined as high-attenuation opacity at the dependent portion of the maxillary sinus measuring ≥ 45 Hounsfield units (HU) as the lower limit of attenuation for clotted blood [15]. Since MHS is a relevant indicator used to detect OFFs on CT scans [5,16], we further classified MHS into the following three CT subtypes: (1) Type 1, high-attenuation opacity mixed with mottled gas (Figure 3A); (2) Type 2, air–fluid level (Figure 3B); and (3) Type 3, full opacification of the sinus (Figure 3C).

The final radiology reports of the head and facial CT scans were interpreted and reviewed by a third radiologist (L.K.H.) who has 8 years of experience with traumatic imaging and were considered the reference standard. This reviewer was aware of the clinical findings and injury-specific data.

### 2.4. Clinical Data Assessment

The clinical data of each patient during the initial presentation at the emergency department were compiled from the electronic medical records using a standard format and included age, sex, reported initial loss of consciousness (ILOC), consciousness and associated Glasgow Coma Scale (GCS) score, mechanisms of head trauma, and pertinent physical findings of the head and face [17,18].

### 2.5. Statistical Analysis

The clinical and demographic characteristics of the patients are summarized as percentages (categorical variables) or the mean and standard deviation (SD) (continuous variables). The comparisons of the categorical variables between the two groups were performed using Fisher’s exact test. The comparisons of the continuous variables between the two groups were performed using independent Student’s *t*-tests. The aforementioned significant factors in the univariate analysis were further analyzed via adjusted multivariate logistic regression models as covariables to determine the independent risk factors of OFFs. The specificity, sensitivity, positive predictive value, and negative predictive value were calculated for the total and each subtype of MHS. Statistical significance was defined as *p* < 0.05 (two-tailed). The statistical analysis was performed using SPSS v.23.0 (Statistical Product and Service Solutions, Chicago, IL, USA).

## 3. Results

### 3.1. Comparison of Patients with and without OFFs

During the 3-year study period, a total of 3534 head trauma patients who received both head and facial CT scans were enrolled in the current study. The 198 patients (5.6%) exhibiting OFFs with or without other coexisting facial fractures were defined as the OFF group, while the other 3336 patients without OFFs observed in their CT images were defined as the non-OFF group. The clinical characteristics and demographics of the patients with and without OFFs are compared in Table 1. The differences between the groups in the univariate analysis included age, sex, ILOC, injury mechanism, GCS score, and physical findings at the emergency department. Furthermore, the multivariate logistic regression analyses revealed that a younger age (odds ratio (OR), 0.98; 95% confidence interval (CI), 0.97–0.99; *p* < 0.001), being male (OR, 1.64; 95% CI, 1.14–2.37; *p* = 0.008), falling from an elevation of more than two meters (OR, 3.22; 95% CI, 1.38–7.54; *p* = 0.007), motorcycle collision (OR, 1.72; 95% CI, 1.17–2.52; *p* = 0.006), and positive physical findings with blepharohematoma (OR, 14.78; 95% CI, 9.94–21.98; *p* < 0.001), facial wounds (OR, 6.68; 95% CI, 4.02–11.11; *p* < 0.001), and epistaxis (OR, 2.47; 95% CI, 1.39–4.39; *p* = 0.002), were independent clinical risk factors of OFFs in head trauma patients. 

The comparison of the head CT variables between the patients with and without OFFs is shown in Table 2. The findings exclusively observed in the patients with OFFs include orbital floor discontinuity (91.4%), gas bubbles entrapped between floor fragments (78.8%), and orbital fat herniation into the maxillary sinus (15.7%). Using these three variables, 187 cases of OFFs (94.4%) were identified by head CT scans alone. In addition, ICH, skull fractures, inferior extraconal emphysema, and MHS were also significantly more commonly observed among patients with OFFs (*p* < 0.001).

### 3.2. Diagnostic Performance of MHS

The sensitivity, specificity, positive predictive value, and negative predictive value of MHS and its three subtypes were evaluated in detail (Figure 4). While MHS exhibited a very high negative predictive value (99.7%) for excluding OFFs, type 1 MHS had the highest positive predictive value (69.5%) for detecting OFFs compared to total MHS and the other two subtypes.

A subset imaging analysis for head trauma patients with MHS was conducted to investigate the correlation between subtypes of MHS and OFFs. Patients with non-orbital floor fractures which may result in MHS (sole maxillary bone or medial orbital wall, or both) were excluded from the non-OFF group. The comparisons of head CT variables between the OFF (*n* = 188) and non-OFF (*n* = 102) groups are shown in Table 3. The OFF group more commonly exhibited type 1 MHS, while types 2 and 3 MHS were more frequently observed in the non-OFF group. In the multivariate logistic regression analyses, after adjusting for other CT variables, only type 1 MHS remained an independent risk factor of OFFs in these patients (OR, 47.50; 95% CI, 8.26–273.05; *p* < 0.001).

## 4. Discussion

In the current study, concomitant orbital fractures were observed in 285 (8.1%) of head trauma patients and OFFs accounted for 70% of these cases. Our results are consistent with the results published by Exadaktylos et al. [10], who also reported that the orbital floor was the most commonly injured site of orbital fractures associated with head trauma. In a report by Foletti et al. that used a finite element model of the human orbit [19], the orbital floor was shown to be the most fragile site in blunt trauma. Furthermore, the correlation of simultaneous craniocerebral injuries and infraorbital rim fractures in blunt trauma was demonstrated by a finite element analysis study [20]. Due to the potential risks of concomitant OFFs and the resulting possible ocular complications in head trauma, early and correct diagnosis of OFFs is necessary during the initial evaluation of head-injured patients at emergency departments. 

The presence of clinical signs usually prompts the initiation of the surveillance of OFFs at emergency departments. Consistent with the reports of a previous study [10], blepharohematoma was a common clinical sign related to orbital fractures in our study population. However, the absence of grossly visible signs cannot rule out OFFs, especially in the pediatric population with white-eye blowout fracture. In this population, only a careful ophthalmic motility examination can uncover OFFs due to the paucity of external signs of injury [21]. Additionally, Exadaktylos et al. suggested the use of clinical symptoms as indicators of the need for additional orbital CT scans in head trauma patients to detect concomitant orbital fractures based on a high negative predictive value of 96.2%. However, a relatively higher asymptomatic rate of 13.8% (4 of 29) was noted in those with isolated OFFs [10]. Furthermore, many factors may contribute to the difficulty of ophthalmologic assessment for the detection of OFFs, such as the lack of ophthalmologists during the initial evaluation at emergency departments and patients arriving unconscious or intubated and unable to cooperate during the ophthalmologic examination.

Given that OFFs may be undetected due to reliance on associated physical indicators and that head CT is the first-line imaging modality for head trauma, in this study, we systemically analyzed the specific findings of OFFs on head CT scans in head trauma patients. The results of the current study showed that orbital floor discontinuity, gas bubbles entrapped between floor fragments, inferior extraconal emphysema, orbital fat herniation into the maxillary sinus, and MHS were significantly more commonly observed in head trauma patients complicated with OFFs. This may suggest that these CT findings are indicators of OFFs. Using the orbitomeatal line (OML) as the imaging reference line in head CT [22], the orbital floor is not completely parallel to the OML [23]. Therefore, orbital floor discontinuity may be identified using axial head CT images. This finding was observed in 91.4% of the patients with OFFs in our study population. In addition, air in the paranasal sinuses enters the orbit through a pathway formed by disrupted sinus mucosa, fractured sinuses and orbital walls, which may be further trapped in the periorbital spaces when the orbital soft tissue or fractured orbital wall act as a one-way valve to block the exit of air [24,25]. Air from the paranasal sinus first enters the adjacent extraconal orbital segment [26]. Therefore, CT findings of gas bubbles entrapped between floor fragments and inferior extraconal emphysema may be observed in the presence of OFFs as demonstrated in 78.8% and 49.5% of our study population, respectively. 

The clear sinus sign has been proposed to exclude paranasal sinus wall fractures in blunt facial trauma patients if there is a lack of free fluid within the associated sinuses on CT images [16]. In accordance with the results of prior studies investigating fractures contiguous with the maxillary sinus [27,28], our study data also suggested that the absence of MHS had a very high negative predictive value (99.7%) for excluding OFFs, whereas the presence of MHS had a relatively low positive predictive value (48.8%) for detecting OFFs. Therefore, we further classified MHS into three subtypes based on the CT findings to investigate their diagnostic value in detecting OFFs. As CT findings of the air–fluid level and complete opacification of the paranasal sinus are frequently observed in patients with rhinosinusitis [29,30], the higher frequency of type 2 and type 3 MHS shown in our study population without OFFs may be related to the preexisting non-hemorrhagic sinus inflammatory disease. Notably, type 1 MHS not only had the highest positive predictive value of approximately 70% for detecting OFFs but was also the only independent CT indicator of OFFs after adjusting for the other CT variables among all head trauma patients with MHS. We hypothesize that because OFFs with interrupted maxillary mucoperiosteal lining contain a profuse vascular supply, a mixture of blood clot and sinus air may present as a high-attenuation opacity containing mottled gas as shown by type 1 MHS. Thus, this finding could be indicative of OFFs. Although the positive predictive value of type 1 MHS is still unsatisfactory because it can also be identified in patients with maxillary sinus or medial orbital wall fractures, our data are not inferior to the results of a prior study using facial CT scans with multiplanar reformatted images [27]. Taken together, head CT scans are important due to their high negative predictive value to obviate unnecessary facial CT in head trauma patients without MHS. In the current study, the average total amount of radiation incidence (dose–length product, DLP) in our patients was 944.27 ± 191.30 mGy-cm on head CT and 293.34 ± 26.33 mGy-cm on facial CT, respectively. Therefore, approximately 30% of the radiation dose could be reduced with the use of MHS as an indicator of OFFs. For head trauma patients with MHS—especially those with type 1 MHS—in addition to head CT and detailed clinical workups for the detection of OFFs, a dedicated facial CT scan is recommended and should be considered in patients without definitive head CT evidence of OFFs. 

Our study had several limitations. First, this study adopted a retrospective design with a patient sample recruited from a single level II trauma center, and the reconstructed CT images with a relatively greater slice thickness used in our institution may lead to underdetection of subtle orbital fractures, especially fissure fractures. Thus, compared to a previous report [10], a relatively lower incidence of orbital fractures and lower injury severities were observed in our study. Second, the presence of extraconal emphysema and MHS may be correlated with the size and type of the orbital fracture [27,28]. In patients with smaller, nondisplaced, or minimally displaced fractures, these CT features may be absent and therefore have limited diagnostic performance in detecting OFFs. Third, the CT variables related to OFFs may be potentially overlooked on routine head CT scans with greater slice thicknesses and increments than facial CT scans. The diagnostic accuracy could be reduced if the images are interpreted by first-line emergency physicians alone or at institutions without on-site radiologists. A head CT protocol with multiplanar reformatted images in the evaluation of head trauma could be considered to facilitate the diagnosis of OFFs without the requirement of additional radiation exposure. Fourth, while MHS is a relevant radiological finding that aids in the diagnosis of OFFs, the maxillary sinus may not be included in the scanning range of routine head CT scans, which could further limit their diagnostic performance in detecting OFFs. Finally, MHS could be demonstrated in THI patients without OFFs but with other facial fractures. Further prospective and multicenter studies using head CT protocols with multiplanar reconstructed images and thinner slice thicknesses are needed to clarify these issues.

## 5. Conclusions

In summary, with a high negative predictive value in the absence of MHS, conventional head CT scans performed in head trauma patients provide discriminatory evidence for excluding concomitant OFFs and therefore obviate unnecessary facial CT. Identifying other findings related to infraorbital injuries on head CT may further aid the diagnosis of OFFs. To avoid unnecessary additional radiation exposure, head CT may serve as a first-line screening tool for the detection of OFFs in head trauma patients. Dedicated facial CT scans may be reserved for those with clinically or radiologically suspicious OFFs to determine the most appropriate management and further surgical planning. 

## Figures and Tables

**Figure 1 jcm-08-01852-f001:**
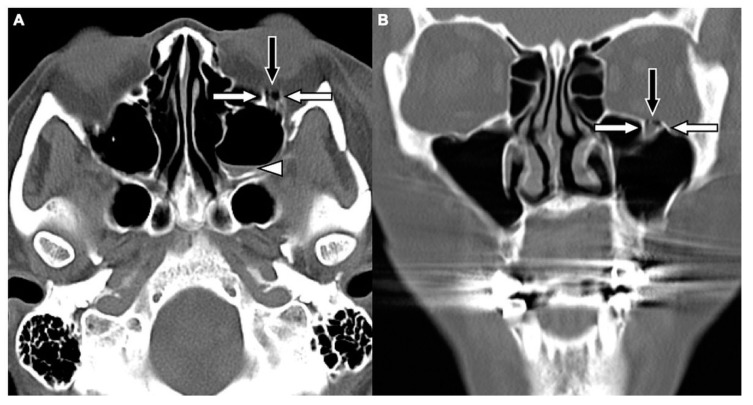
CT images of a 29-year-old male with assault-related head trauma and a concomitant left orbital floor fracture (OFF). An axial head CT image (**A**) and the corresponding coronal facial CT image (**B**) show gas bubbles (black arrow) entrapped between the discontinuous floor fragments (white arrows). Left type 2 maxillary hemosinus (MHS) (arrowhead) and a left zygomatic fracture are also noted in (A).

**Figure 2 jcm-08-01852-f002:**
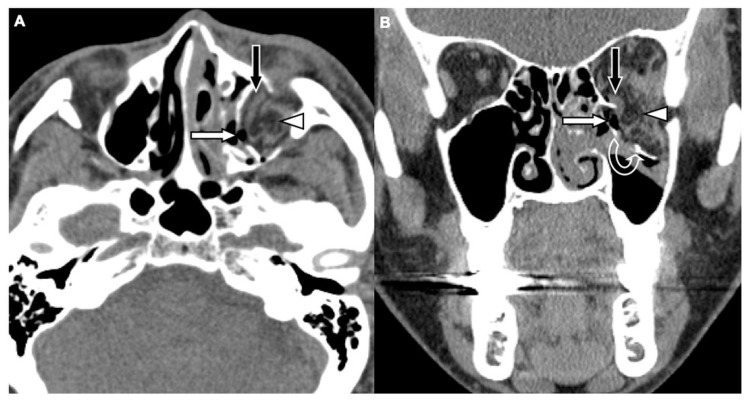
CT images of a 62-year-old female with motorcycle crash-related head trauma and a concomitant left OFF. An axial head CT image (**A**) and the corresponding coronal facial CT image (**B**) show inferior extraconal emphysema (straight white arrow) beneath the inferior rectus muscle (black arrow), orbital fat herniation into the maxillary sinus (arrowhead), and the depressed fragments of fractured orbital floor (open curved arrow).

**Figure 3 jcm-08-01852-f003:**
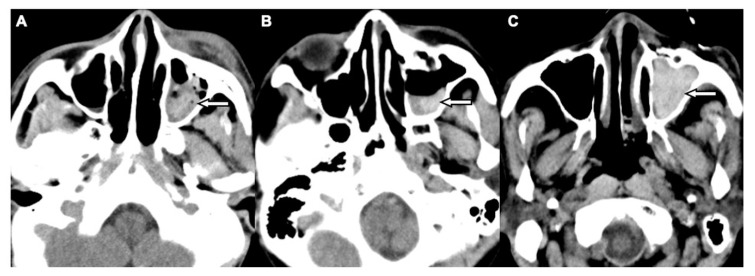
CT images showing three patterns of MHS related to OFFs associated with head trauma. (**A**) Type 1: high-attenuation opacity mixed with mottled gas in the left maxillary sinus (arrow) in a 52-year-old male with motorcycle crash-related head trauma. (**B**) Type 2: air–fluid level in the left maxillary sinus (arrow) in a 26-year-old female with motor vehicle collision-related head trauma. (**C**) Type 3: full opacification of the left maxillary sinus (arrow) in a 79-year-old male with fall accident-related head trauma.

**Figure 4 jcm-08-01852-f004:**
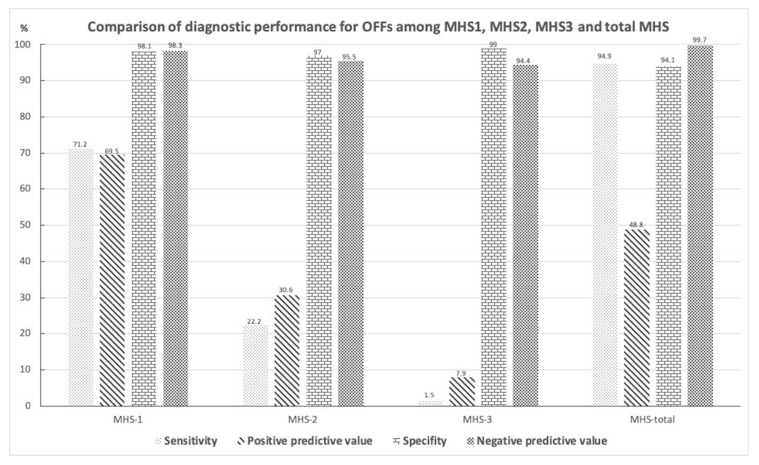
Diagnostic performance of the total and the three subtypes of MHS.

**Table 1 jcm-08-01852-t001:** Comparisons of clinical and demographic variables between head trauma patients with and without OFFs.

Variables	With OFFs (*n* = 198)	Without OFFs (*n* = 3336)	*p*-Value	Odds Ratio of OFF ^a^ (95% CI)
**General demographics**				
Age (years)	46.8 ± 21.5	58.6 ± 21.0	<0.001 ^b^	0.98 (0.97–0.99)
Male (*n*, %)	137 (69.2)	1810 (54.3)	<0.001 ^c^	1.64 (1.14–2.37)
**Clinical information**				
ILOC (*n*, %)	98 (49.5)	985 (29.5)	<0.001^c^	NS
Specific mechanism of injury (*n*, %)				
Fall from elevation	11 (5.6)	96 (2.9)	0.033 ^c^	3.22 (1.38–7.54)
Motorcycle collision	116 (58.6)	880 (26.4)	<0.001^c^	1.72 (1.17–2.52)
**Evaluation in emergency department**				
GCS at ED (scale)	13.7 ± 3.0	14.6 ± 1.7	<0.001 ^b^	NS
LOC (GCS < 14) (*n*, %)	47 (23.7)	259 (7.8)	<0.001 ^c^	NS
Deep coma (GCS < 8) (*n*, %)	19 (9.6)	81 (2.4)	<0.001 ^c^	NS
Physical findings (*n*, %)				
Blepharohematoma	159 (80.3)	535 (16.0)	<0.001 ^c^	14.78 (9.94–21.98)
Facial wound	179 (90.4)	1139 (34.1)	<0.001 ^c^	6.68 (4.02–11.11)
Epistaxis	26 (13.1)	95 (2.8)	<0.001^c^	2.47 (1.39–4.39)

Values represent the mean ± SD. Abbreviations: CI, confidence interval; GCS, Glasgow Coma Scale; ILOC, initial loss of consciousness; IQR, interquartile range; LOC, loss of consciousness; NS, not significant; OFFs, orbital floor fractures. ^a^ Multivariate regression analysis. ^b^ Independent Student’s *t*-test. ^c^ Fisher’s exact test.

**Table 2 jcm-08-01852-t002:** Comparisons of head CT variables between head trauma patients with and without OFFs.

Variables	With OFFs (*n* = 198)	Without OFFs (*n* = 3336)	*p*-Value ^a^
CT variables related to the cranium (*n*, %)			
ICH	52 (26.3)	463 (13.9)	<0.001
Skull fracture	37 (18.7)	150 (4.5)	<0.001
CT variables related to OFFs (*n*, %)			
Orbital floor discontinuity	181 (91.4)	0 (0)	<0.001
Gas bubbles entrapped between floor fragments	156 (78.8)	0 (0)	<0.001
Orbital fat herniation into the maxillary sinus	31 (15.7)	0 (0)	<0.001
Inferior extraconal emphysema	98 (49.5)	15 (0.4)	<0.001
MHS	188 (94.9)	197 (5.9)	<0.001

Abbreviations: CT, computed tomography; ICH, intracranial hemorrhage; MHS, maxillary hemosinus; OFFs, orbital floor fractures. ^a^ Fisher’s exact test.

**Table 3 jcm-08-01852-t003:** Comparisons of head CT variables related to OFFs in 290 head trauma patients with MHS.

CT Variables	With OFFs (*n* = 188)	Without OFFs ^a^ (*n* = 102)	*p*-Value ^b^	Odds Ratio ^c^ (95% CI)
MHS type 1	141 (75.0)	7 (6.9)	<0.001	47.50 (8.26–273.05)
MHS type 2	44 (23.4)	63 (61.8)	<0.001	-
MHS type 3	3 (1.6)	32 (31.4)	<0.001	-
Orbital floor discontinuity	172 (91.5)	0 (0)	<0.001	NS
Gas bubbles entrapped between floor fragments	149 (79.3)	0 (0)	<0.001	NS
Orbital fat herniation into the maxillary sinus	27 (14.4)	0 (0)	<0.001	NS
Inferior extraconal emphysema	94 (50.0)	0 (0)	<0.001	NS

Abbreviations: CI, confidence interval; CT, computed tomography; MHS, maxillary hemosinus; NS, not significant; OFFs, orbital floor fractures. ^a^ Patients with fractures involving sole maxillary bone or medial orbital wall, or both in the non-OFF group were excluded. ^b^ Fisher’s exact test. ^c^ Multivariate regression analysis.

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
