# Peer review of "Evaluation of Concomitant Orbital Floor Fractures in Patients with Head Trauma Using Conventional Head CT Scan: A Retrospective Study at a Level II Trauma Center"

_jcm, 2019, doi:10.3390/jcm8111852_

Round 1

Reviewer 1 Report

This is a well-written and well-conducted study but if the great advantage of this study is to prove that orbital floor fracture are associated with radiologic signs found with just a head CT scan (axial plane) such as maxillary hemosinus type 1 (high-attenuation opacity and mottled gas), this is a too long article for just this finding.

The authors should shorten some parts (for instance in the results section the demographic characteristics are not really the topic of the article) and the discussion : is it really necessary to assess that "..patients with clinical signs highly suspicious of facial fracture, the scanning fields of head CT should be extended to include the facial region..." ? This is just common sense, isn't it ?

Reviewer 2 Report

The study is well written and it is nothing more a descriptive epidemiologic retrospective view of patient with concomitant head injury and orbital fractures (OFFs). The overall message is unclear. The authors describe some radiological signs associated with OFFs in payients with a traumatic head injury. The orbit is usually captured scanning the brain when a THI occur becouse of the posterior cranial base is lower than the floor of the orbit. For the reader the message is unclear. The would like  to focus on the importance of orbital scanning? Or on the association of radiological orbital  signs to THI? Reading the results, it is clear that there is a correlation between the energy of the trauma with the presence of OFFs. Moreover clinical singns in table 2 are more correlated to OFFs. So this evaluation should be considered as first line in choosing a Head CT scan or Head an face CT scan. Other Authors suggested clinical signs as discriminatory indicators of additional face CT scan. Finally the study is well written, but the message is unclear and not original. I would like to suggest to change the title in a decriptive study  taking more consideration on clinic signs and trauma's injury energy level.  I cannot see NOVEL interpretations or new considerations about MHS. Due to the large number of patients the Authors may reconsider submission changing the target of the study. I would like to recommned Evaluation of concomitant........using conventional herad CT scan: a retrospective study at level I trauma center. 

I recommend the publication after major revisions.

Reviewer 3 Report

In the manuscript, the authors have described about a new modality in the evaluation of concomitant orbital floor fractures in patients with traumatic head injury using conventional head CT scanning. I have the following comments regarding the manuscript:

There are some errors in spacing, missed commas and font size. The authors should correct all of them before submission of the revised manuscript. I disagree with the comment in page 2, lines 46-9. First, for ophthalmologists, cataract is not a severe disease because this can be treated with surgery. Second, how can taking orbital CT once cause cataract and carcinogenesis in the orbit? I could not understand the imaging analysis. The authors mentioned in page 3, lines 83-4 that 2 radiologists reviewed “only the head CT” taken from the foramen magnum to the skull vertex. How did these radiologists judge changes in the orbits and maxillary sinuses? I could not understand the grading scale of hemorrhage in the maxillary sinus. A major flaw of this study was inclusion of mixed cases with maxillary/zygomatic bone and medial orbital wall fractures. Sole maxillary bone or medial orbital wall fractures also cause hemorrhage in the maxillary sinuses. Therefore, they should exclude such patients from “without OFFs” group. In addition, they should mention the inclusion of patients with impure orbital floor fracture in the participants section. I could not understand why the authors stuck to orbital floor fracture. They had reasoned that among orbital fractures, the orbital floor is the most commonly involved wall and this can cause enophthalmos. However, the fracture site is different according to age and the medial wall is the most commonly involved site in old patients. Moreover, patients with orbital floor fracture have less risk of enophthalmos than those with medial orbital wall fracture. To begin with, the authors reviewed coronal facial CT images, on which we can easily diagnose orbital floor fracture without the need for checking hemorrhage in the maxillary sinuses and orbital emphysema. I would recommend them not to use unfamiliar abbreviations, such as THI, ED, PE, PPV, and NPV because these cause poor readability. In page 3, in the caption of Figure 1, should write the statement “Left type 2 MHS (arrowhead) and a left zygomatic fracture are also noted in Figure 1(A)”, as this finding is not seen in Figure 1(B). Pages 3-4. Kindly, do proper labelling in the respective CT images as “A”, “B” or “C”. Page 9, 2nd Kindly, split the 2nd sentence into two, so that it will lead to better readability.

Round 2

Reviewer 3 Report

The authors had revised their paper well.